# Factors associated with food consumption and dietary diversity among infants aged 6–18 months in Ashanti Region, Ghana

Godwin Opoku Agyemang[1]*, Samuel Selorm Attu[1], Reginald Adjetey Annan[1], Satoru Okonogi[2], Takeshi Sakura[2], Odeafo Asamoah-Boakye[1]

1 Department of Biochemistry and Biotechnology, College of Science, Kwame Nkrumah University of Science and Technology, PMB, Kumasi, Ghana, 2 Department of Agricultural and Resource Economics, the University of Tokyo, Tokyo, Japan

* godwinopokuagyemang6@gmail.com

**Data Availability Statement:** Data cannot be shared publicly due to the fact that it contains sensitive identifying information. However, data are

## Abstract

### Background

Studies have reported a poor diet quality among children in Ghana and other developing countries. Inadequate dietary diversity among children may lead to deficiencies in micronutrient intake necessary for growth and other cognitive functions. Understanding factors associated with infants' diverse diets is a key step to promoting adequate infant and young child nutrition. This study sought to determine the factors associated with food consumption and dietary diversity (DD) among infants.

### Methods

In this cross-sectional study among 1503 mothers-infant (aged 6–18 months) pairs from rural, urban, and peri-urban districts of Ashanti Region, factors associated with food consumption and DD were evaluated. The FAO 18-food group DD questionnaire was used to determine previous food group intake, while a structured questionnaire was used to capture data on the mother's socio-demographic parameters and child morbidity. Data were analyzed using descriptive statistics, the Chi-square test, and binary logistic regression to compare mothers and infants who had adequate and inadequate DDS of 9 out of 18 food groups and the predictors of dietary diversity.

### Results

About 64.7% of the infants did not meet the minimum dietary diversity. Over two-third (77.4%) of the children consumed maize porridge the previous day. Foods which were less consumed included vegetables (35%), animal milk (38.9%), and meat (organ 14%, any meat 26%). The mean food group intake from 18 food groups was 7.0, and the majority (64.7%) did not meet the recommended 9 food group intake. Significantly more younger children (6–11 months) (74%, p<0.001) compared with older children (12–18 months) (52.5%) did not meet the minimum DDS. Also compared with the older children, the younger ones had above two times increased odds of inadequate DD (OR = 2.5, p<0.001, 95% CI =

available from the Committee on Human Research, Publication, and Ethics (CHRPE), the ethics board of the School of Medicine and Dentistry of the Kwame Nkrumah University of Science and Technology (KNUST), and Komfo Anokye Teaching Hospital (KATH), Kumasi, Ghana (email: chrpe. knust.kath@gmail.com) for researchers who meet the criteria for access to confidential data.

**Funding:** The authors received no specific funding for this work.

**Competing interests:** The authors have declared that no competing interests exist.

1.4–4.4). When controlled for gender, children from peri-urban areas (OR = 5.2, p = 0.260, 95% CI = 0.2–93.2) and rural areas (OR = 1.8, p = 0.650, 95% CI = 0.2–9.3) had increased odds of lower DD than urban children. Children of unemployed caregivers had an increased odds of low DD (OR = 2.3 p<0.001, 95% CI 1.7–3.2) compared with children of employed caregivers. Finally, children of caregivers with better nutrition knowledge (nine correct answers from 12 questions) had lower odds of having lower dietary diversity (adjusted OR = 0.9, p = 0.85, 95%CI = 0.5–1.6) than those with less knowledge.

## Conclusions

Low DD was common among infants and associated with infants age, caregivers' areas of residence, employment status, and level of nutrition education. Children who did not meet the minimum dietary diversity were not fed particular foods such as vegetables, animal milk, and organ meat. Proper maternal nutrition education and feeding practices targeting age-specific needs and community livelihood support systems are necessary to improve dietary diversity of infants.

## 1. Introduction

The concept of optimum feeding is a key determinant of growth and development among infants and young children [1]. According to the United Nations Children's Fund, the age of 6–23 months is considered the critical period in the first 1000 days of life because infants transition from feeding on breastmilk only to the inclusion of complementary foods [2]. Suboptimal infant feeding practice is a major public health concern among children, particularly in less developed communities [3]. Poor feeding practices among infant and young children are on the rise [4]. According the World Health Organization (2018), only less than 25% of infants between 6–23 months consume the recommended diverse diet [5]. Several studies have reported a poor dietary diversity among infants in developing countries [4]. A high prevalence of minimum dietary diversity has been reported in South Africa (43.9%), Tanzania (26%), and Ethiopia (23.3%) [6]. A study conducted by Frempong and Annim revealed that 47% of children in Ghana met the minimum dietary diversity [7]. Studies show that complementary foods given to young children in Ghana are mostly foods of plant origin and tend to have reduced amounts of nutrients [8]. Majority of infants in sub-Saharan Africa (SSA) are fed with cereal based supplemental feeds which are mostly nutritionally inadequate to support rapid growth [9].

Dietary diversity plays an important role in the health of infants. The World Health Organization recommends the provision of a diverse diet to ensure adequate intake of energy, vitamins, proteins, fats, and minerals through complementary feeding for infants and young children aged 6 months and above [10]. *"Minimum dietary diversity is defined as consumption of food from four or more out of seven food groups for higher dietary quality and to meet daily energy and nutrient requirements of the seven recommended food groups namely: grains, roots and tubers; legumes and nuts; dairy products; flesh foods (meat, fish, poultry and organ meat), eggs; vitamin-A rich fruits and vegetables; other fruits and vegetables"* Minimum dietary diversity is one of the eight indicators of infant and young child feeding practices as proposed by the World Health Organization [11]. Increased variety of dietary intake is usually proportional to improved diet quality, characterized by more animal-source foods than a less diversified

one [12]. Consistent intake of diverse food sources is essential for adequately meeting nutrient demands [13]. The consumption of a diverse diet during infancy is relevant for maintaining proper health, growth and development of infants and children [14].

Inadequate dietary diversity among children leads to deficiencies in micronutrient intake necessary for growth and other cognitive functions [15]. A large population study reported that the majority of children who did not meet the recommended nutrient intake, including folate, vitamin E, zinc, and fiber, performed poorly in a cognitive test [16]. Consumption of inadequate diverse diet among infants may lead to undernutrition, risk of infections and severe illness [17]. Evidence suggests that poor feeding practices, including an insufficient quantity of complementary and poor-quality foods, are linked with childhood malnutrition [14]. According to the UNICEF, 3.1 million children die from undernutrition yearly. Poor feeding practices at the ages of 6–23 months are the leading cause of undernutrition and other infant morbidities in developing countries, including Ghana [18].

Several studies have reported a positive association between dietary diversity and micronutrient intake, employment status, household food security, socioeconomic status and nutrition knowledge [19, 20]. Families with higher income status tend to have consume a diverse diet which results in a positive nutritional status of their infants and young children [20]. Another study which accessed the nutrition knowledge and complementary feeding practices among 1400 mothers in the Northern part of Ghana showed a positive relationship between maternal nutrition knowledge and appropriate feeding practices [21]. According to the study, women who had higher nutrition-related knowledge and attitude were almost twice likely to feed their children with appropriate diet. Poor dietary diversity among infants is reported to be associated with child's age, geographical region and cultural traditional barriers. Understanding factors associated with food consumption and dietary diversity among a large population of infants is a critical step which will be necessary in designing appropriate interventions. This will help improve the proportion of children who are fed with diversified diet in sub-Saharan Africa. However, in Ghana and other sub-Saharan Africa countries, fewer studies have focused on the dietary diversity and food consumption of infants aged between 6 and 18 months who have transitioned from breastmilk only to include complementary foods. Also, very few studies in Ghana have reported on factors associated with poor diet quality in infants aged 6 and 18 months. In this study, we used a large population and primary data to report on factors associated with food consumption and dietary diversity in infants aged between 6 and 18 months. Due to the various food variations in the different regions of Ghana, conducting such a study in Ashanti Region, the second most populous region in Ghana with diverse ethnicity and socioeconomic status will provide a general representation of food consumption and dietary diversity among infants and further serve as a basis for policy implementation.

## 2. Materials and methods

### 2.1 Study design

Data for this manuscript was generated from a larger study entitled "Experimental Evidence of Effectiveness and Efficiency of Market Based Approach to Improvement of Infant Nutrition in Ghana". The study sought to evaluate the effect of KOKOPLUS infant food supplement on children's nutritional status as well as evaluating the effect of information about children's current nutritional status on demand for KOKOPLUS. The study was a community-based non-controlled intervention study among mother-infant (6–18) months old which was conducted in the Child Welfare Clinics (CWC) in Asokore Mampong Municipality and Bosomtwe District of the Ashanti Region of Ghana. Ashanti Region is the second most populous city in Ghana noted for its ethnic diversity. Also noted for its farming activities, the region has viable

lands which support the cultivation of several crops, fruits and vegetables. It is also noted for economic and other trading activities. Therefore, it is believed that with the availability and access to the different food groups in the region, genuine factors associated with dietary diversity will be determined. Also, the presence of different ethnic groups in the region could be a general representation of the entire country. Asokore Mampong Municipality is one of the newly carved districts and forms part of the region's 43 Metropolitan, Municipal, and District Assemblies (MMDAs). It is an urban community with an estimated population of 372, 222 inhabitants, according to the Population and Housing Census of 2010. The land has an area of 23.9 km$^2$ and a density of 15,558/km$^2$. The township of Asokore Mampong can be described as a society of mixed ethnicity. The dominant group is the Akans (40.9%), followed by people from the Northern Region, (36.7%). On the other hand, Bosomtwe District is one of the 43 Districts in the Ashanti Region, located in the Middle Belt Authority of Ghana. In the 2010 census, Bosomtwe District had a population of 93, 910. A majority of the district is made up of Ashanti with smaller proportions of the ethnic group like Ga, Dagombas, and Fante. About 36.3% of the district's population is employed in agriculture. The study captured a baseline data between April and May, 2019 after which six months monthly follow up was conducted between June and November 2019. An end of intervention was also conducted between November and December, 2019. This manuscript was written from data generated in the baseline study. The baseline captured data on breastfeeding and complementary feeding knowledge and practices, anthropometric data and household socioeconomic status.

## 2.2 Study population and sampling method

A total of 1503 mothers and their indexed infants aged 6–18 months were recruited for the study. A total of 28 CWCs were selected from a total of 40. All mothers whose infants were aged 6–18 months and were attending the selected CWC were recruited during the four weeks of the recruitment period. The mothers were conveniently sampled during their routine visits to the community weighing centers (CWC) in their communities. Mothers who had twin infants and those whose children were outside the age bracket of the study were excluded. The date of birth (day/month/year) of infants, which was recorded in the child's weight records books, was used to determine the infants ages. A list of mother-infant (6–18 months) pairs from the community weighing centers (CWC) was generated, and every child within the selected age range was eligible for inclusion. Data was collected using digital tablets on which the questionnaire had been uploaded to the SurveyBe software application. Data was captured in four parts, including participants' socio-demographic characteristics, maternal nutrition knowledge on breastfeeding, and complementary feeding practices.

## 2.3 Independent variable

The age, education, occupation, ethnicity, and religion of the mothers were assessed under the socio-demographic section. The past and current feeding practices of participants were assessed based on the WHO recommendations on infant and young child feeding practices (IYCF). The complementary feeding practices, such as timely initiation of complementary foods, number of complementary foods, food consistency, meal frequency, dietary diversity, use of micronutrient supplements, and feeding behavior during illness and recovery, were assessed. The WHO's key recommendations for breastfeeding and complementary feeding were used in the assessment of maternal nutrition knowledge and practices. A total of thirteen (13) questions were adopted. Questions were adapted from the WHO indicators for infant feeding. A correct score of "1" was given for a single question, while a "0" was given for an incorrect answer. The total marks were obtained by the summation of the "1" scored for all

questions. A participant who scored more than 9 out of the 13 questions, which is equivalent to 70%, was then classified as having adequate nutrition knowledge on breastfeeding, while others were otherwise classified as having inadequate nutrition knowledge. This grading was in line with a similar study conducted in Ghana by [14]. Respondents were asked when they started giving anything apart from breastmilk to a child. The amount of food given to the child in the next 24 hours was estimated. The consumption of foods from specific food groups was also assessed. Food groups included grains, roots, tubers, dairy, eggs, legumes, nuts, fruits, vegetables, and fleshy foods. This was based on the mother's ability to recall foods given to her child in the past 24 hours.

## 2.4 Dependent variable

Caregivers were asked to select from a list of 18 food items the number of foods consumed by their children in the past 24 hours [22]. These eighteen (18) foods were captured from the eight food groups proposed by the Food and Agriculture Organization (2010), for screening for the minimum dietary diversity. For each food group consumed by the infant in the past 24 hours before data collection, a score of "1" was given, and "0" was given for non-consumed foods. The dietary diversity score was determined from the sum of all consumed food groups on the 18-food list. We defined the dietary diversity of the children by using a similar definition for the minimum dietary diversity of women (MDD-W). In the MDD-W, 10 food groups are used to determine the dietary diversity of women and pregnant women. Consuming five or more food groups is classified as adequate dietary diversity, while consuming less than five is classified as inadequate dietary diversity. Similarly, this study used an expanded 18-food list. A cut-off of nine or more food groups was used to define 'met dietary diversity, while consuming less than nine was defined as 'not met dietary diversity.

## 2.5 Ethics

A general ethical approval was obtained for the larger study from the Committee on Human Research Publication and Ethics (CHRPE), School of Medical Science, Kwame Nkrumah University of Science and Technology, and Health Director of Health Services at Asokore Mampong (reference: CHRPE/AP/098/19). This current manuscript forms part of the larger study. Ethical approval was given in the name of Miss Nancy Elizabeth Tandoh who was part of the six postgraduate students who worked on different areas of this larger study. Trained enumerators explained the purpose of the study to the mothers with the help of nurses working at the CWCs. After each data collection, the data was well reviewed and screened for completeness by the supervisors and principal investigators. The purpose of the study was clearly explained to the mothers with the help of nurses working at the CWCs upon recruitment. Before conducting the study, participants were given written consent forms to sign following research ethics regulations.

## 2.6 Data analysis

Data were extracted from the SurveyBe application and uploaded onto the Statistical Package for Social Sciences (SPSS) version 25.0 (SPSS IBM Inc., Chicago, USA) for statistical analysis. Microsoft Excel was used to draw a graph of food group consumption against the percentage of children who consumed the various food groups. A Chi square (fisher's exact test) cross-tabulation analysis was conducted to determine the relationship between child food group consumption and dietary diversity. A chi-squared analysis was also conducted to determine the relationship between sociodemographic variables and dietary diversity. One sample T-test was also used to determine the mean, standard deviation, and error mean for the dietary diversity

score, maternal age, and the relationship between child morbidity, maternal nutrition knowledge, and dietary diversity was also examined using the Chi square (fisher's exact test) crosstabulation. A binary logistic regression analysis was conducted for all relevant significant variables to determine the predictors of poor child dietary diversity. All tests were 2-tailed, and a p-value of < 0.05 was defined as statistically significant.

## 3. Results

Fig 1 shows the percentage distribution of food consumed by the children in the past 24 hours. Among the food groups, the majority of the children (77.4%) consumed porridge. More than half of the children did not consume foods including vegetables, beans, sugary foods, milk, and oils. Meat consumption was very poor among children, as 26% and 74% did not consume organ meat or any other meat.

Fig 2 shows the dietary diversity among the infants. From the figure, dietary diversity was poor among the infants as it can be observed that more than half of the infants (64.7%) did not meet the minimum dietary diversity.

Table 1 shows the relationship between food consumption and the dietary diversity of the children. From the table, children who did not consume some foods did not meet the minimum dietary diversity, and children who consumed most foods the previous day were more likely to meet the minimum dietary diversity. The majority of the children who consumed foods including dark green vegetables, meat, and milk had adequate dietary diversity. A higher proportion of children who did not consume fruits (56.8%, P<0.001) had adequate dietary diversity score compared to those who consumed fruits and did not meet the dietary diversity

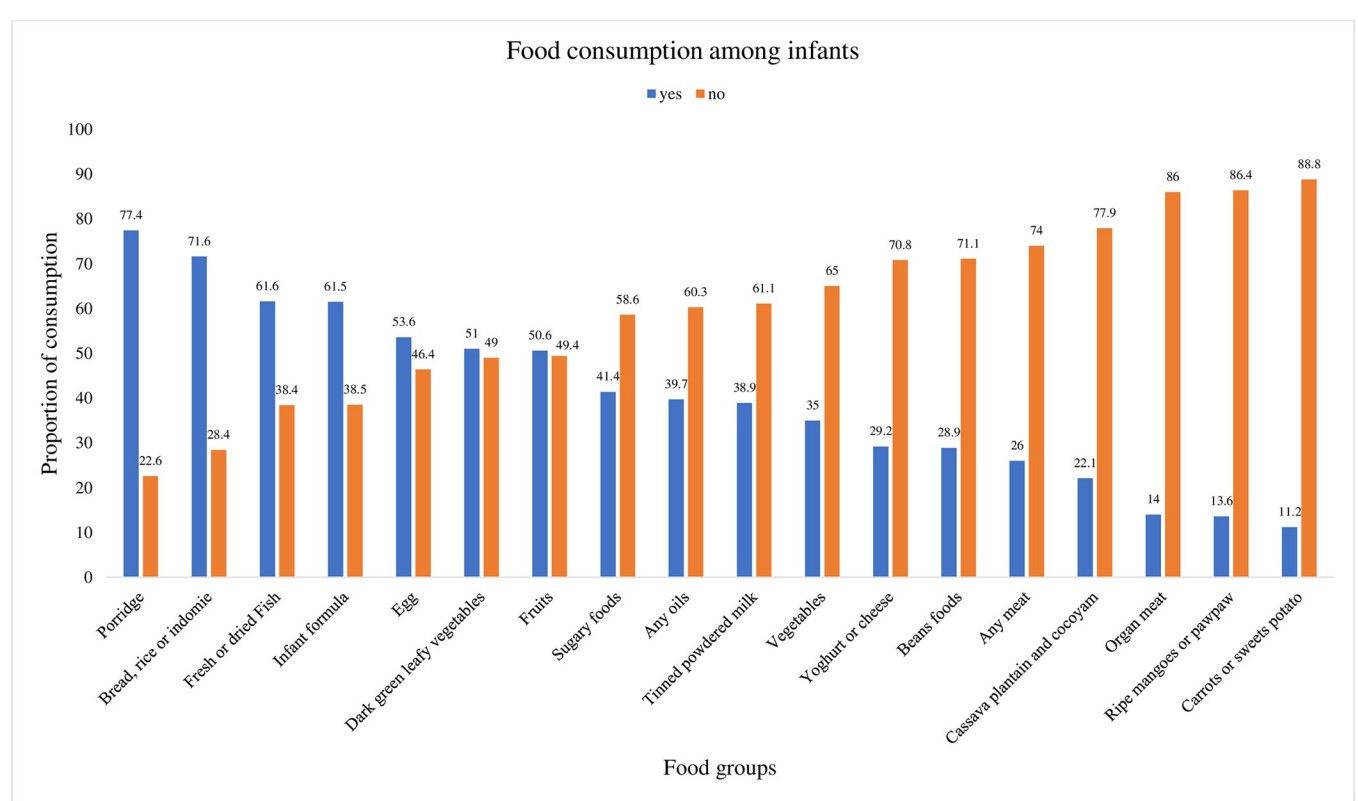

**Fig 1. Distribution of food consumption by children in the past 24 hours.**

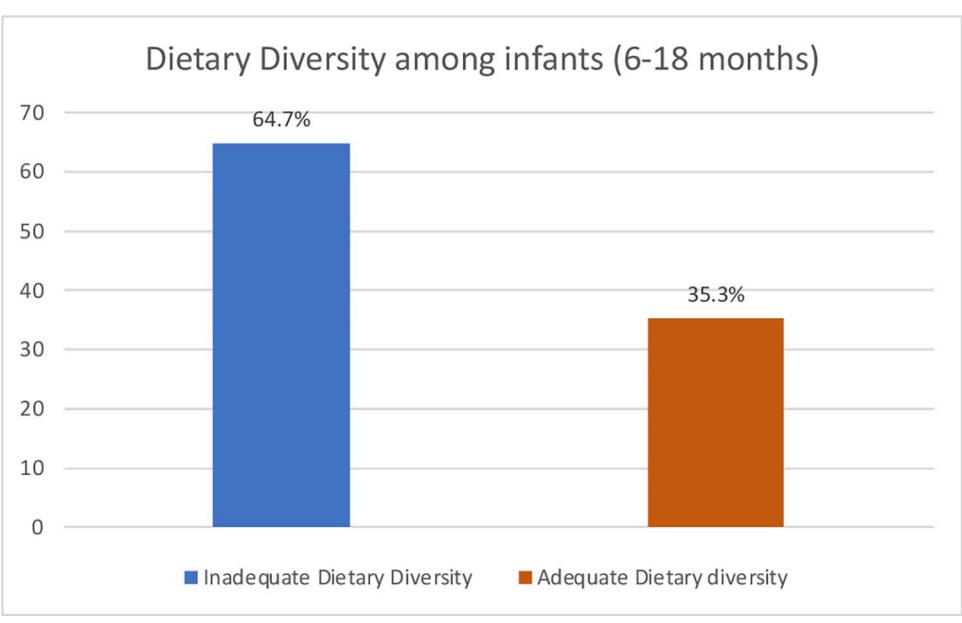

**Fig 2. Dietary diversity among infants between 6–18 months.**

(43.2%). Similar observations can be seen in children who consumed vegetables (62.5%, P<0.001). S1 and S2 Tables represent food group consumption by age groups and the dietary diversity of children. A higher proportion of 12–18 months old children who consumed bread, rice, or noodles (54.5%) met the DD (p<0.001) compared with 6–11 months old children who

**Table 1. Relationship between food consumption and dietary diversity of children.**

| Food consumption | Inadequate DD N (%) | adequate DD N (%) | $X^2$ | P-value |
|---|---|---|---|---|
| Bread, rice, noodles | 564 (53.3) | 494 (46.7) | 212.94 | <0.001* |
| Cassava, plantain, cocoyam | 129 (39.3) | 199 (60.7) | 119.1 | <0.001* |
| Porridge | 682 (59.7) | 460 (40.3) | 55.26 | <0.001* |
| Any oils | 232 (39.4) | 357 (60.6) | 275.46 | <0.001* |
| Sugary foods | 266 (43.3) | 348 (56.7) | 210.84 | <0.001* |
| Infant formula | 295 (54.0) | 251 (46.0) | 43.4 | <0.001* |
| Powdered or fresh animal milk | 254 (42.2) | 336 (57.8) | 213.44 | <0.001* |
| Egg | 364 (45.8) | 431 (54.2) | 270.5 | <0.001* |
| Fish or dried fish | 446 (49.1) | 463 (50.9) | 253.91 | <0.001* |
| Organ meat | 44 (21.2) | 164 (78.8) | 201.3 | <0.001* |
| Any meat | 109 (28.3) | 276 (71.7) | 302.42 | <0.001* |
| Beans foods | 120 (28.1) | 307 (71.9) | 352.84 | <0.001* |
| Yogurt or cheese | 132 (30.6) | 299 (69.4) | 309.94 | <0.001* |
| Fruits | 321 (43.2) | 422 (56.8) | 303.33 | <0.001* |
| Carrots or sweet potato | 31 (18.7) | 135 (81.3) | 173.71 | <0.001* |
| Dark green leafy vegetables | 337 (44.8) | 416 (55.2) | 268.36 | <0.001* |
| Ripe mangoes or pawpaw | 38 (19.0) | 162 (81.0) | 211.84 | <0.001* |
| Vegetables | 196 (37.5) | 326 (62.5) | 261.183 | <0.001* |

Data are reported as frequency (percentage), Chi square ($X^2$) P-values are reported, P-value is significant at p<0.05, DD-Dietary Diversity

consumed bread, rice, noodles (39.0%). Also, children between 12–18 months who consumed porridge (52.5%) the previous day had adequate dietary diversity (p<0.001) as compared to their counterparts between 6–11 months who consumed porridge (31.0%). This can also be seen in the consumption of infant formula, where a higher proportion of older infants who consumed infant formula (63.9%) had adequate dietary diversity compared to the younger adults who consume infant formula (36.9%). A higher proportion of children aged 12–18 months who consumed any meat (76.0%) had adequate dietary diversity as compared to their counterparts who consumed meat (64.3%).

Table 2 shows the relationship between the socio-demographic factors and the recommended dietary diversity score of the participants. The table also shows the means of the

**Table 2. Relationship between the socio-demographic factors and dietary diversity score of infants.**

| Sociodemographic variables | Total N = 1503 | Inadequate DD | Adequate DD | X$^2$ | P-value |
|---|---|---|---|---|---|
| | N (%) | (< 9 food groups)<br>N = 973 (64.7%) | (≥9 food groups)<br>N = 530 (35.3%) | | |
| **Gender of Child** | | | | | |
| Male | 750 (49.9) | 489 (66.4) | 248 (33.6) | 1.7 | 0.212 |
| Female | 753 (50.1) | 467 (63.1) | 273 (36.9) | | |
| **Child's Age (months)** | | | | | |
| 6–11 | 856 (57) | 622 (74.0) | 219 (26.0) | 72.93 | <0.001* |
| 12–18 | 647 (43) | 334 (52.5) | 302 (47.5) | | |
| **Mother's Age** | | | | | |
| < 20 | 634 (46.8) | 400 (64.1) | 224 (35.9) | 0.34 | 0.841 |
| 20–29 | 54 (4) | 32 (60.4) | 21 (39.6) | | |
| 30–45 | 668 (49.2) | 423 (64.4) | 234 (35.6) | | |
| **Resident** | | | | | |
| Rural | 189(12.6) | 153 (81.8) | 34 (18.2) | 70.53 | <0.001* |
| Peri-Urban | 258 (17.2) | 205 (79.8) | 52 (20.2) | | |
| Urban | 1056 (70.3) | 598 (57.9) | 435 (42.1) | | |
| **Maternal education** | | | | | |
| Basic | 298 (20.7) | 188 (64.8) | 102 (35.2) | 0.87 | 0.832 |
| JHS | 750 (52.1) | 487 (65.8) | 253 (34.2) | | |
| Secondary | 283 (19.7) | 183 (66.1) | 94 (33.9) | | |
| Tertiary | 108 (7.5) | 74 (69.8) | 32 (30.2) | | |
| **Employment status** | | | | | |
| Employed | 789 (71) | 469 (61.0) | 300 (39.0) | 32.71 | <0.001* |
| Unemployed | 322 (29) | 252 (79.0) | 67 (21.0) | | |
| **Income status** | | | | | |
| 1000 cedis and more | 56 (12.5) | 38 (67.9) | 18 (32.1) | 0.83 | 0.653 |
| 500–999 cedis | 118 (26.3) | 70 (60.9) | 45 (39.1) | | |
| Less than 500 cedis | 274 (61.2) | 167 (62.1) | 102 (37.9) | | |
| **Variable** | **Mean** | **±mean error (SD)** | | | |
| Child age (months) | 11.0 | 3.3 (0.8) | | | |
| Maternal age | 30.1 | 11.9 (0.3) | | | |
| Income in cedis (¢) | 609.3 | 0.03 (0.6) | | | |
| Mean maternal knowledge | 7.0 | 1.4 (0.2) | | | |

Frequency (percentage) and mean, standard deviation (SD) standard error means are reported, chi-square(X$^2$) values are reported, and p-values are significant at p < 0.05

**Table 3. Child morbidity in the past seven days and dietary diversity status of children.**

| Child morbidity in the past 7 days | Inadequate DD (< 9 food groups) | Adequate DD (≥ 9 food groups) | $X^2$ | P-value |
|---|---|---|---|---|
| **Running Nose** | | | | |
| Yes | 433 (64.4) | 239 (35.6) | 0.03 | 0.871 |
| No | 521 (64.9) | 282 (35.1) | | |
| **Fever** | | | | |
| Yes | 250 (63.0) | 147 (37.0) | 0.69 | 0.425 |
| No | 704 (65.3) | 374 (34.7) | | |
| **Cough** | | | | |
| Yes | 267 (61.9) | 164 (38.1) | 1.98 | 0.168 |
| No | 687 (65.8) | 357 (34.2) | | |
| **Diarrhoea** | | | | |
| Yes | 190 (66.7) | 95 (33.3) | 0.61 | 0.449 |
| No | 764 (64.2) | 426 (35.8) | | |
| **Vomiting** | | | | |
| Yes | 106 (63.1) | 62 (36.9) | 0.20 | 0.668 |
| No | 848 (64.9) | 459 (35.1) | | |

Frequency (percentage), chi-square ($X^2$) P values reported, p-values are significant at p < 0.05

child's age, mother's age, income, and the mean dietary diversity score. The mean age of the children was 11 months. About 65% of the children did not meet the recommended food group intake. A higher proportion of 6–11 months old children (74.0%) had inadequate DD (p<0.001) compared with 12–18 months old children (52.5%). The majority of the children from rural areas (81.8%) could not meet the DD (p<0.001). A greater proportion of children whose mothers were unemployed had inadequate DD (p<0.001).

Table 3 shows the relationship between child morbidity in the past seven days and the dietary diversity status of infants. From the table, there was no significant difference between child morbidity and dietary diversity score, although most of the children who had no illness for the past seven days recorded a low recommended dietary diversity score. However, children who had diarrhea in the previous week (66.7%) were more likely to have inadequate DD compared with those who had no diarrhea (64.2%). Similarly, children who vomited continuously in the previous week (63.1%) were not likely to consume diverse diet as compared to their counterparts who did not vomit (64.9%).

Table 4 shows the relationship between nutrition knowledge and the dietary diversity status of the children. The mean nutrition knowledge of the mothers was 7.0 (p<0.001). Children of caregivers who had poor nutrition knowledge (69.9%, P = 0.023) had low dietary diversity scores. Children whose caregivers knew when to provide them with family food (61.9%) had better dietary diversity scores than children whose caregivers did not know when to start giving family food to their children (69.1, p<0.001). Also, caregivers who knew how many times to give complementary food to their children at 9–12 months had their children record better dietary diversity scores (44.1%, 0.00) as compared to their counterparts who did not know (34.1%, p<0.001). However, majority of children of caregivers who knew the quantity of complementary food to give them rather had poor dietary diversity scores (67.9%, p<0.001).

Table 5 indicates the predictive factors associated with Child's dietary diversity. Children within 6–11 months (adjusted OR = 2.5, p<0.001, 95%CI = 1.4–4.4) had higher odds of having low dietary diversity compared to the children between 12–18 months. Children who lived in rural (unadjusted OR = 3.2, p<0.001, 95%CI = 2.2–4.8) and peri-urban centers (unadjusted OR = 2.8, p<0.001, 95%CI = 2.0–3.9) had higher odds of having a low dietary diversity as

**Table 4. Relationship between maternal nutrition knowledge and dietary diversity status of infants.**

| Mothers' nutrition knowledge | Inadequate DD | Adequate DD | X2 | P-value |
|---|---|---|---|---|
| Good knowledge | 130 (59.9) | 87 (40.1) | 3.21 | 0.023* |
| Satisfactory Knowledge | 775 (65.3) | 412 (34.7) | | |
| Poor knowledge | 51 (69.9) | 22 (30.1) | | |
| **When should a woman breastfeed after birth** | | | | |
| Correct | 822 (65.4) | 435 (34.6) | 1.64 | 0.221 |
| Wrong | 134 (60.9) | 86 (39.1) | | |
| **How long should you exclusively breastfeed without adding any drink or food** | | | | |
| Correct | 913 (64.7) | 499 (35.3) | 0.2 | 0.63 |
| Wrong | 27 (61.4) | 17 (38.6) | | |
| **How long should you breastfeed before stopping or weaning** | | | | |
| Correct | 705 (63.9) | 399 (36.1) | 1.44 | 0.231 |
| Wrong | 251 (67.3) | 122 (32.7) | | |
| **When should a woman start to give water after birth** | | | | |
| Wrong | 954 (64.7) | 521 (35.3) | 0.3 | 0.601 |
| Correct | 549 (61.2) | 445 (38.8) | 0.3 | |
| **When should a woman give food after birth?** | | | | |
| Correct | 835 (64.8) | 454 (35.2) | 0.1 | 0.742 |
| Wrong | 117 (63.6) | 67 (36.4) | | |
| **When should the child start eating family food?** | | | | |
| Correct | 551 (61.9) | 339 (38.1) | 7.66 | <0.001* |
| Wrong | 386 (69.1) | 173 (30.9) | | |
| **How much complementary food should one give at 6 months** | | | | |
| Correct | 597 (67.9) | 282 (32.1) | 9.59 | <0.001* |
| Wrong | 358 (60.1) | 238 (39.9) | | |
| **How much complementary food should one give a 7–8 months** | | | | |
| Correct | 260 (62.2) | 158 (37.8) | 1.62 | 0.203 |
| Wrong | 694 (65.7) | 362 (34.3) | | |
| **How much complementary food should one give at 9–12 months** | | | | |
| Correct | 251 (57.8) | 183 (42.2) | 0.12 | 0.743 |
| Wrong | 297 (56.7) | 227 (43.3) | | |
| **How many times should you give complementary food at 6–8 months** | | | | |
| Correct | 683 (64.7) | 373 (35.3) | 0.13 | 0.567 |
| Wrong | 271 (64.8) | 147 (35.2) | | |
| **How many times should you give complementary food at 9–12 months** | | | | |
| Correct | 104 (55.9) | 82 (44.1) | 7.11 | 0.009* |
| Wrong | 847 (65.9) | 438 (34.1) | | |
| **Should the child be fed different kinds of food or the same type** | | | | |
| Correct | 924 (64.3) | 512 (35.7) | 3.81 | 0.061 |
| Wrong | 31 (79.5) | 8 (20.5) | | |

Frequency (percentage), chi-square P values reported, p-values are significant at p < 0.05

compared to children who lived in urban centers. Children whose mothers were unemployed (unadjusted OR = 2.3, p<0.001, 95%CI = 1.7–3.2) had increased odds of having low dietary diversity as compared to their counterparts whose mothers were employed. Children whose mothers had adequate nutrition knowledge had fewer odds of having lower dietary diversity (Adjusted OR = 0.9, p = 0.851, 95%CI = 0.5–1.6).

**Table 5. Predictors of factors associated with child's dietary diversity.**

| Predictable variable | Adjusted OR (95%CI) | P-value | Unadjusted OR (95%CI) | P-value |
|---|---|---|---|---|
| **Age** | | | | |
| 12–18 months | 1 | | 1 | |
| 6–11 months | 2.5 (1.4–4.4) | <0.001 | 2.5 (2.0–3.1) | <0.001* |
| **Residence** | | | | |
| Urban | 1 | | 1 | |
| Rural | 1.8 (0.1–29.1) | 0.651 | 3.2 (2.2–4.8) | <0.001* |
| Peri–urban | 5.2 (0.2–93.2) | 0.261 | 2.8 (2.0–3.9) | <0.001* |
| **Employment status** | | | | |
| Employed | 1 | | 1 | |
| Unemployed | 1.1 (0.1–10.6) | 0.925 | 2.3 (1.7–3.2) | <0.001* |
| **Maternal Nutrition knowledge;** | | | | |
| 1. When should a woman give the family food | 1.8 (1.0–3.2) | 0.025 | 1.3 (1.0–1.7) | <0.001* |
| 2. how much complementary food should a child eat at 6months | 0.9 (0.5–1.6) | 0.851 | 1.4 (1.1–1.7) | <0.001* |
| 3. How many times should you feed a child at 9 months to 12 months | 0.9 (0.4–2.2) | 0.935 | 1.5 (1.1–2.0) | <0.001* |

OR- Odds ratio, AOR- Adjusted odds ratio, CI- Confidence interval, and p-values are significant at p<0.05, adjusted for gender

## 4. Discussion

This study identified factors associated with food consumption and dietary diversity among infants in Ghana. Severally implemented policies in Ghana including the National Nutrition Policy has been directed at ensuring the healthy growth and development of infants and young children. However, poor feeding practices are reported among children and young infants in Ghana. From our study, the majority of the children consumed porridge (77.4%) and bread, rice, and noodles (71.6%). A similar study in Ghana reported 81% consumption of bread, noodles, and other grains among children [23]. On the contrary, the intake of milk, vegetables, and meat was very low among the children. Similarly, a study among infants between 6–23 months in rural communities in Ethiopia reported low consumption of animal products including meat and milk [24]. Our study found that more than 6 in 10 (64.7%) children had poor-quality diets. This represents a very low level of dietary diversity among the children. Our findings are in line with a study conducted in Ghana by Bandoh and Kenu, which reported a prevalence of 54% of children not meeting the minimum dietary diversity [25]. A higher prevalence of low dietary diversity was reported among infants between 6–23 months in Tanzania [6]. This indicates that the majority of children in the country are likely to be prone to micronutrient deficiencies. Poor feeding among infants is the cause of infant morbidity and mortality. Therefore, necessary nutrition interventions must be put in place to tackle such issues. A popular approach to curtailing this problem has been the free distribution of nutritionally dense complementary foods or supplements [26].

The findings revealed that some socio-demographic factors were associated with child dietary diversity. Child age was found to be associated with dietary diversity, as 74% of the children who did not meet the minimum dietary diversity were between 6–11 months. According to the study, children in this age group were twice at risk of having a low dietary diversity as compared to their counterparts between 12–18 months. This is consistent with other studies that reported a minimum dietary diversity score among children who were between 6–11 months old in Ghana [27]. The reason is due to the preparedness of older infants to accept food in various forms and their high familiarity with certain foods as compared to younger infants. According to Mura Paroche et al., exposure of an infant to a variety of diets at a

young age is efficient in ensuring the liking and consumption of both exposed and new foods [28].

Our study also reported how child residence was a predictor of dietary diversity. About 41% of the children who met the minimum dietary diversity were living in urban centers. Children in rural areas had three times the odds of having a lower dietary diversity than their counterparts in urban centers. Similar findings were seen in a study conducted by [29] among Ghanaian adolescents, where rural dwellers had higher odds of inadequate dietary diversity compared to those in urban centers. A study in Nepal reported a higher prevalence of inadequate dietary diversity among children living in developed areas compared to less developed areas [30]. This can also be seen in other studies that reported a more diversified food intake among urban residents as compared to rural residents [31, 32]. Urban centers are characterized by food access and high economic status, which may explain the decreased risk of having low dietary diversity as compared to rural areas [33]. Majority of rural areas in Ghana are noted for farming of foodstuffs, fruits and vegetables, however, such cultivation is usually meant for commercial purposes to generate income for the family. Though was not found in the study, certain cultural practices prominent in the rural areas might account for the poor dietary diversity among the children. About 37% of pregnant women in a study reported one or more food practices influenced by local beliefs and cultural taboos [34]. Such poor cultural beliefs should not be encouraged since it could hinder the consumption of some micronutrients among infants.

According to our study, maternal employment status is also a predictor of infant dietary diversity. Unemployed women had twice the odds of having a lower dietary diversity score for their children as compared to the other women who were employed. Similar issues were seen in a study conducted by Tampah-Naah, where mothers in Ghana who had challenges accessing complementary food items was due to a lack of income or insufficient financial endowment [35]. In a similar study which reported a low dietary diversity among infants, a higher proportion of mothers could not afford a diverse diet because they had difficulty obtaining a decent income [36]. Interventional policies to engage women in livelihood support, as well as other financial support, are critical to addressing this issue. In 2008, the Livelihood Empowerment Against Poverty was initiated to established in Ghana to improve basic household consumption and nutrition, and increase access to welfare services. Strengthening this policy will provide women access to better livelihood where pregnant women can afford nutrient-dense foods.

According to the study, child morbidity was not a predictor of dietary diversity; however, other studies have reported the contrary. Findings from a study by [37] found that child morbidity was a contributing factor to low dietary diversity. Their study also reported low dietary diversity among children suffering from diarrhea. This is, however, consistent with our study, where children who had diarrhea in the previous week (66.7%) were more likely not to meet the DD compared with those who had no diarrhea (64.2%).

Though certain studies have identified a positive association between maternal education level and child dietary diversity [19, 38] our study found no association between maternal educational level and dietary diversity. Similarly, level of education among breastfeeding mothers in India was not associated with breastfeeding practices [39]. However, from our findings, children from mothers who had relatively adequate nutrition knowledge had higher chances of meeting the minimum dietary diversity as compared to the other group. Maternal nutrition knowledge has been established as a predictor of infant feeding practices and nutritional status. A similar observation is seen in a study in Ghana by [40], where households with caregivers who had adequate nutrition knowledge and attitude scores had more diverse diets compared to those with little or no nutrition knowledge. Proper maternal nutrition education

interventions are therefore critical to solving such situations. Currently, the Ghana Health Service recommends the intake of a four-star diet for a healthy nutrition. Through the health care systems, pregnant and breastfeeding mothers as part of the antenatal and post-natal services are educated on the consumption of the four-star diet. Empowering pregnant women with nutrition knowledge will help them make better choices which will eventually ensure a healthy nutrition outcome.

## 4.1 Strengths and limitations of the study

Our study used participants from different socioeconomic setting to determine the factors that were associated with food consumption and dietary diversity among the children as compared to previous studies in Ghana and Sub-Sahara Africa. However, our study was not without limitations. Other factors which can also affect child dietary diversity such as food availability and accessibility and cultural and ethnic factors were not assessed in our study. Moreover, the use of the 24-hour dietary recall is not a true indicator of dietary diversity though it is widely used. However, the 24-hour dietary recall is less subject to recall errors, less cumbersome for the respondent and conforms to the recall time period used in many dietary diversity studies.

## 5. Conclusions

The study found that infants had poor-quality diet. The majority of the infants also consumed fewer food groups from the 18-food group list. Our study has shown that the factors associated with poor dietary diversity among the infants were multifactorial ranging from age-specific, occupation, nutritional knowledge of mother and residence of infants. Based on the findings in this study, we suggest that proper maternal nutrition education and feeding practices targeting age-specific needs and community livelihood support systems are necessary to improve dietary diversity of infants.

## Supporting information

**S1 Table. Food group consumption by age groups and dietary diversity of infants.**
(PDF)

**S2 Table. Regression analysis showing the predictors of infant dietary diversity.**
(PDF)

## Acknowledgments

We thank the directors of health and health workers in the health centers where the study took place and most of all the mothers and infants who participated in the study.

## Author Contributions

**Conceptualization:** Godwin Opoku Agyemang, Reginald Adjetey Annan.

**Data curation:** Godwin Opoku Agyemang, Samuel Selorm Attu, Reginald Adjetey Annan, Odeafo Asamoah-Boakye.

**Formal analysis:** Godwin Opoku Agyemang, Reginald Adjetey Annan, Odeafo Asamoah-Boakye.

**Funding acquisition:** Reginald Adjetey Annan.

**Investigation:** Godwin Opoku Agyemang, Reginald Adjetey Annan.

**Methodology:** Godwin Opoku Agyemang, Reginald Adjetey Annan.

**Project administration:** Godwin Opoku Agyemang, Samuel Selorm Attu, Reginald Adjetey Annan.

**Resources:** Godwin Opoku Agyemang, Reginald Adjetey Annan.

**Software:** Godwin Opoku Agyemang, Reginald Adjetey Annan, Odeafo Asamoah-Boakye.

**Supervision:** Reginald Adjetey Annan, Satoru Okonogi, Takeshi Sakura.

**Writing – original draft:** Godwin Opoku Agyemang.

**Writing – review & editing:** Godwin Opoku Agyemang, Samuel Selorm Attu, Reginald Adjetey Annan, Satoru Okonogi, Takeshi Sakura, Odeafo Asamoah-Boakye.

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
