## [Decision Letter · Decision Letter 0]

25 Jun 2023

PONE-D-23-17357FACTORS ASSOCIATED WITH FOOD CONSUMPTION AND DIETARY DIVERSITY AMONG INFANTS AGED 6-18 MONTHS IN ASHANTI REGION GHANAPLOS ONE

Dear Dr. Opoku Agyemang,

Thank you for submitting your manuscript to PLOS ONE. After careful consideration, we feel that it has merit but does not fully meet PLOS ONE’s publication criteria as it currently stands. Therefore, we invite you to submit a revised version of the manuscript that addresses the points raised during the review process.

We look forward to receiving your revised manuscript.

Kind regards,

Pradip Chouhan

Academic Editor

PLOS ONE

Journal Requirements:

"NO"

Reviewers' comments:

Reviewer's Responses to Questions

**Comments to the Author**

1. Is the manuscript technically sound, and do the data support the conclusions?

Reviewer #1: Yes

Reviewer #2: Yes

Reviewer #3: No

2. Has the statistical analysis been performed appropriately and rigorously? 

Reviewer #1: Yes

Reviewer #2: Yes

Reviewer #3: No

3. Have the authors made all data underlying the findings in their manuscript fully available?

Reviewer #1: Yes

Reviewer #2: Yes

Reviewer #3: Yes

4. Is the manuscript presented in an intelligible fashion and written in standard English?

Reviewer #1: No

Reviewer #2: Yes

Reviewer #3: No

5. Review Comments to the Author

Reviewer #1: Title: Factors Associated with Food Consumption and Dietary Diversity Among Infants Aged 6-18 Months in Ashanti Region Ghana

Authors have highlighted an interesting and significant issues through this manuscript.

1) Abstract:

Well Written. Need to rewrite the background part of the abstract. Try to address objective of the study in abstract part.

2) Introduction: Need to improve.

a) The research gap is unclear. Overall, the introduction is unclear as to what the authors want to say. It is better to rewrite based on i) what the issue is, ii) why it is important, iii) what the literature says, iv) what the research gap and question in this study are, and v) how this study could contribute to the literature.

b) The concept, methods and results are cohesive and scientifically sound. If I would nit-pick, there are some issues with grammar. However, the manuscript can be understood regardless.

c) Yet, to be of interest for readers outside Ghana, too, the paper should also refer to some studies in the international literature (studies done on the children of others countries) on child feeding practice, to highlight the gaps and unanswered questions, the methodologies used etc. Authors may cite following literature on breastfeeding practice in India

Ghosh, P., Rohatgi, P., Bose, K. (2022). Determinants of Time-Trends in Exclusivity and Continuation of Breastfeeding in India: An Investigation from the National Family Health Survey. Social Science and Medicine, 292, 1-10. https://doi.org/10.1016/j.socscimed.2021.114604

3) Methodology: Well written

4) Results: Well Written

5) Discussion: Need to improve

a) The Discussion section tells a coherent story of what was discovered and the implications for Ghana. I suggest the authors to shed light on those results that may be generalizable (outside the Ghana context). This section should also reflect the contribution of the study to the international literature in the study area.

b) In this section authors have already summarized the process, the results, and the overall purpose of this study.

c) One paragraph/part should answer questions about the limitations and potential flaws or shortcomings of this study. One paragraph should focus on the successes of the study.

d) Authors can also compare the results of different methods and assess which was more fruitful and why. Discuss the implications of this research and compare it to the results of other studies. Authors can cite related studies to show how your study compares (in addition to the introduction).

6) Conclusion: The conclusion offers you a chance to briefly present how your work advances the field from the present state of knowledge. Make a proper justification for your work here and indicate extensions and wider implications of the study.

Reviewer #2: The paper titled "FACTORS ASSOCIATED WITH FOOD CONSUMPTION AND DIETARY DIVERSITY AMONG INFANTS AGED 6-18 MONTHS IN THE ASHANTI REGION, GHANA" presents an intriguing study. With the revisions suggested below, I am confident that this paper can be strengthened and transformed into a captivating published article.

1. The rationale behind selecting the Ashanti Region in Ghana as the study area needs to be explained and justified by the author.

2. The research paper lacks novelty, and it is necessary to incorporate this in the end of the introduction section of the manuscript.

3. Check spelling/writing whole manuscript. Several errors are available in the manuscript (IN ASHANTI REGION GHANA…. (77.4%)… (35%)

4. Please rewrite the whole “materials and methods” section. I am unable to understand how many independent and dependent variables there are. Do not make it complicated. Try to simplify.

5. The author refers to the prevalence of child dietary diversity, but unfortunately, I couldn't locate the specific table for this.

6. Figure 1 is not clearly visible.

7. How did you select the sample for your study? Could you please provide details about the procedure used for sample selection?

8. Please include detailed analytical techniques (formulas, descriptions etc.) for "Dominance analysis of relative contribution" and "Binary logistic regression" in the appendix for the reader.

9. I couldn't find any information regarding existing policies or policy enhancement strategies in the discussion section of this study related to food consumption and dietary diversity among infants aged 6-18 months in Ghana.

10. Other necessary factors that impact food consumption and dietary diversity among infants aged 6-18 months should be considered. These factors include maternal and family support, parental feeding practices (such as pressure to eat), food availability and accessibility, and cultural and ethnic factors. Please add.

11. What are the strength and limitations of the study? Please mention.

12. In the discussion section, it is unnecessary to restate the result of the study. “(unadjusted OR=2.3, p. Please remove.

13. What interventions and strategies can be implemented to enhance food consumption and dietary diversity among infants aged 6-18 months in Ghana? Please recommend this in the conclusion section of the manuscript.

Reviewer #3: 1. In introduction part, dietary diversity and it's relation with impaired growth need to be clearly mentioned. Dietary diversity and balanced meal are same?

2. In methodology section

1. Why the study participants taken between 6-18 months and not 6 to 23 months which is standard.

2. 24 hours recall method is not a proper estimation of dietary diversity specially here 9 food groups were taken as to maintain minimum dietary diversity which may not reflect the true diet habit for the participant.

3. In result section,

1. table 1: The dietary diversity presence calculated by scoring 18 food groups intake, also test of significance applied taking these food groups as independent variables. Multicollinearity will be present for statistical test, thus can not be applied.

2. None of the table mentioned the total number of sample size (n) in title.

3. the knowledge of mother regarding diet were shown in open ended but options given as dichotomous. The statements need to be modified.

4. The title and description didn't match in table 5. In title it is mentioned as dietary diversity. also in methodology as presence of dietary diversity. But in binary regression the impression was made as presence of lower dietary diversity which should be mentioned in data analysis section and also the title needed to be changed.

5. In supplementary table 2, how the test of significance was applied is not clear as the table had 3 variables.

6. PLOS authors have the option to publish the peer review history of their article (what does this mean?). If published, this will include your full peer review and any attached files.

Reviewer #1: No

Reviewer #2: No

Reviewer #3: No

---

## [Author Response · Author response to Decision Letter 0]

7 Aug 2023

Please a rebuttal letter which contains the response to all the reviewers and editor's comments have been attached as a file to the attachments.

---

## [Decision Letter · Decision Letter 1]

19 Sep 2023

PONE-D-23-17357R1FACTORS ASSOCIATED WITH FOOD CONSUMPTION AND DIETARY DIVERSITY AMONG INFANTS AGED 6-18 MONTHS IN ASHANTI REGION, GHANAPLOS ONE

Dear Dr. Godwin Opoku Agyemang,

Thank you for submitting your manuscript to PLOS ONE. After careful consideration, we feel that it has merit but does not fully meet PLOS ONE’s publication criteria as it currently stands. Therefore, we invite you to submit a revised version of the manuscript that addresses the points raised during the review process.

We look forward to receiving your revised manuscript.

Kind regards,

Pradip Chouhan

Academic Editor

PLOS ONE

Journal Requirements:

**Comments to the Author**

1. If the authors have adequately addressed your comments raised in a previous round of review and you feel that this manuscript is now acceptable for publication, you may indicate that here to bypass the “Comments to the Author” section, enter your conflict of interest statement in the “Confidential to Editor” section, and submit your "Accept" recommendation.

Reviewer #1: All comments have been addressed

Reviewer #2: All comments have been addressed

2. Is the manuscript technically sound, and do the data support the conclusions?

Reviewer #1: Yes

Reviewer #2: Yes

3. Has the statistical analysis been performed appropriately and rigorously? 

Reviewer #1: Yes

Reviewer #2: Yes

4. Have the authors made all data underlying the findings in their manuscript fully available?

Reviewer #1: Yes

Reviewer #2: Yes

5. Is the manuscript presented in an intelligible fashion and written in standard English?

Reviewer #1: Yes

Reviewer #2: Yes

6. Review Comments to the Author

Reviewer #1: Authors address the issues that I have suggested previously. The author has revised all areas that is possible for this study.

Reviewer #2: 1. In the Materials and Methods section of the manuscript, keep only "Dependent variable:" and "Independent variable:" labeled as point. and remove the word "Data collection." Remove "socio-demographic," "complementary feeding knowledge," "complementary feeding practices," "child morbidity," and "dietary diversity" after "Dependent variable:" and "Independent variable:

2. In the "Strengths and Limitations of the study" section, remove the phrases "a very large sample size and" and "with a relatively smaller sample size" from the first sentence.

3. Check and very ole literatures should be replaced by the updated ones.

4. Include DOI or link to each reference.

7. PLOS authors have the option to publish the peer review history of their article (what does this mean?). If published, this will include your full peer review and any attached files.

Reviewer #1: **Yes: **Dr. Pritam Ghosh

Reviewer #2: No

---

## [Author Response · Author response to Decision Letter 1]

3 Oct 2023

Please all comments raised by the editor and reviewer have been addressed accordingly and attached as 'Response to Reviewers'.

---

## [Editor Report · Decision Letter 2]

10 Oct 2023

PONE-D-23-17357R2FACTORS ASSOCIATED WITH FOOD CONSUMPTION AND DIETARY DIVERSITY AMONG INFANTS AGED 6-18 MONTHS IN ASHANTI REGION, GHANAPLOS ONE

Dear Dr. Opoku Agyemang,

Thank you for submitting your manuscript to PLOS ONE. After careful consideration, we feel that it has merit but does not fully meet PLOS ONE’s publication criteria as it currently stands. Therefore, we invite you to submit a revised version of the manuscript that addresses the points raised during the review process.

ACADEMIC EDITOR: Please insert comments here and delete this placeholder text when finished. Be sure to:

In the conclusion of the abstract , kindly add the implications of your study and/or recommendations. 

Insert line numbers in the document. 

I have included additional documents in the attached file. 

We look forward to receiving your revised manuscript.

Kind regards,

Sandra Boatemaa Kushitor, Ph.D.

Academic Editor

PLOS ONE
---

## [Author Response · Author response to Decision Letter 2]

3 Nov 2023

Please all suggestions from the academic editor have been addressed.

---

## [Editor Report · Decision Letter 3]

10 Nov 2023

FACTORS ASSOCIATED WITH FOOD CONSUMPTION AND DIETARY DIVERSITY AMONG INFANTS AGED 6-18 MONTHS IN ASHANTI REGION, GHANA

PONE-D-23-17357R3

Dear Dr. Agyemang, 

We’re pleased to inform you that your manuscript has been judged scientifically suitable for publication and will be formally accepted for publication once it meets all outstanding technical requirements.

Kind regards,

Sandra Boatemaa Kushitor, Ph.D.

Academic Editor

PLOS ONE
---

## [Editor Report · Acceptance letter]

20 Nov 2023

PONE-D-23-17357R3 

FACTORS ASSOCIATED WITH FOOD CONSUMPTION AND DIETARY DIVERSITY AMONG INFANTS AGED 6-18 MONTHS IN ASHANTI REGION, GHANA 

Dear Dr. Opoku Agyemang:

I'm pleased to inform you that your manuscript has been deemed suitable for publication in PLOS ONE. Congratulations! Your manuscript is now with our production department. 

Kind regards, 

on behalf of

Dr. Sandra Boatemaa Kushitor 

Academic Editor

PLOS ONE